# ZERO-SHOT RETRIEVAL WITH SEARCH AGENTS AND HYBRID ENVIRONMENTS

## ABSTRACT

Learning to search is the task of building artificial agents that learn to autonomously use a search box to find information. So far, it has been shown that current language models can learn symbolic query reformulation policies, in combination with traditional term-based retrieval, but fall short of outperforming neural retrievers. We extend the previous learning to search setup to a hybrid environment, which accepts discrete query refinement operations, after a first-pass retrieval step via a dual encoder. Experiments on the BEIR task show that search agents, trained via behavioral cloning, outperform the underlying search system based on a combined dual encoder retriever and cross encoder reranker. Furthermore, we find that simple heuristic Hybrid Retrieval Environments (HRE) can improve baseline performance by several nDCG points. The search agent based on HRE (HARE) matches state-of-the-art performance, balanced in both zero-shot and in-domain evaluations, via interpretable actions, and at twice the speed.

## 1 INTRODUCTION

Transformer-based dual encoders for retrieval, and cross encoders for ranking (cf. e.g., Karpukhin et al. (2020); Nogueira & Cho (2019)), have redefined the architecture of choice for information search systems. However, sparse term-based inverted index architectures still hold their ground, especially in out-of-domain, or *zero-shot*, evaluations. On the one hand, neural encoders are prone to overfitting on training artifacts (Lewis et al., 2021). On the other, sparse methods such as BM25 (Robertson & Zaragoza, 2009) may implicitly benefit from term-overlap bias in common datasets (Ren et al., 2022). Recent work has explored various forms of dense-sparse hybrid combinations, to strike better variance-bias tradeoffs (Khattab & Zaharia, 2020; Formal et al., 2021b; Chen et al., 2021; 2022).

Rosa et al. (2022) evaluate a simple hybrid design which takes out the dual encoder altogether and simply applies a cross encoder reranker to the top documents retrieved by BM25. This solution couples the better generalization properties of BM25 and high-capacity cross encoders, setting the current SOTA on BEIR by reranking 1000 documents. However, this is not very practical as reranking is computationally expensive. More fundamentally, it is not easy to get insights on why results are reranked the way they are. Thus, the implicit opacity of neural systems is not addressed.

We propose a novel hybrid design based on the Learning to Search (L2S) framework (Adolphs et al., 2022). In L2S the goal is to learn a search agent that autonomously interacts with the retrieval environment to improve results. By iteratively leveraging pseudo *relevance feedback* (Rocchio, 1971), and language models' *understanding*, search agents engage in a goal-oriented traversal of the answer space, which aspires to model the ability to 'rabbit hole' of human searchers (Russell, 2019). The framework is also appealing because of the interpretability of the agent's actions.

Adolphs et al. (2022) show that search agents based on large language models can learn effective symbolic search policies, in a sparse retrieval environment, but fail to outperform neural retrievers. We extend L2S to a dense-sparse hybrid agent-environment framework structured as follows. The environment relies on both a state-of-the-art dual encoder, GTR (Ni et al., 2021), and BM25 which separately access the document collection. Results are combined and sorted by means of a transformer cross encoder reranker (Jagerman et al., 2022). We call this a Hybrid Retrieval Environment (HRE). Our search agent (HARE) interacts with HRE by iteratively refining the query via search operators, and aggregating the best results. HARE matches state-of-the-art results on the

BEIR dataset (Thakur et al., 2021) by reranking a one order of magnitude less documents than the SOTA system (Rosa et al., 2022), reducing latency by 50%. Furthermore, HARE does not sacrifice in-domain performance. The agent's actions are interpretable and dig deep in HRE's rankings.

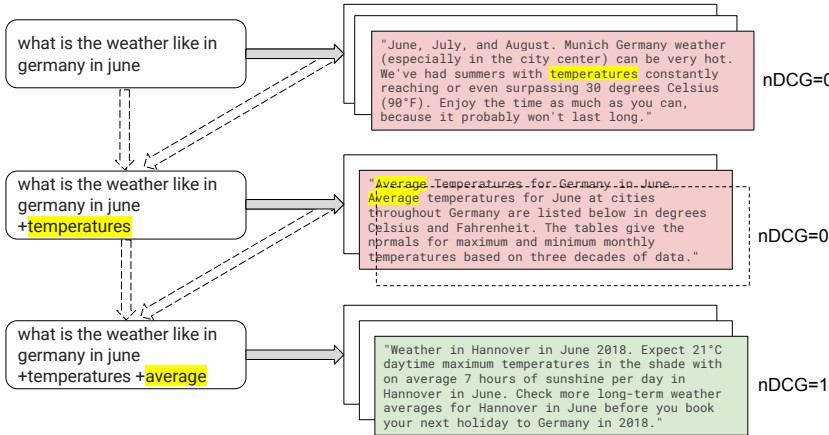

Figure 1: Sequential query refinements combining pseudo relevance feedback and search operators.

Figure 1 shows an example of a search session performed by the HARE search agent applying structured query refinement operations. The agent adds two successive filtering actions to the query 'what is the weather like in germany in june' (data from MS MARCO (Nguyen et al., 2016)). In the first step it restricts results to documents containing the term 'temperatures', which occurs in the first set of results. In the second step, results are further limited to documents containing the term 'average'. This fully solves the original query by producing an nDCG@10 score of 1.

## 2 RELATED WORK

Classic retrieval systems such as BM25 (Robertson & Zaragoza, 2009) use term frequency statistics to determine the relevancy of a document for a given query. Recently, neural retrieval models have become more popular and started to outperform classic systems on multiple search tasks. Karpukhin et al. (2020) use a dual-encoder setup based on BERT (Devlin et al., 2019), called DPR, to encode query and documents separately and use maximum inner product search (Shrivastava & Li, 2014) to find a match. They use this model to improve recall and answer quality for multiple open-domain question-answer datasets. Large encoder-decoder models such as T5 (Raffel et al., 2020) are now preferred as the basis for dual encoding as they outperform encoders-only retrievers (Ni et al., 2021).

It has been observed that dense retrievers can fail to catch trivial query-document syntactic matches involving n-grams or entities (Karpukhin et al., 2020; Xiong et al., 2021; Sciavolino et al., 2021). ColBERT (Khattab & Zaharia, 2020) gives more importance to individual terms by means of a *late interaction* multi-vector representation framework, in which individual term embeddings are accounted in the computation of the query-document relevance score. This is expensive as many more vectors need to be stored for each indexed object. ColBERTv2 (Santhanam et al., 2022) combines late interaction with more lightweight token representations. SPLADE (Formal et al., 2021b) is another approach that relies on sparse representations, this time induced from a transformer's masked heads. SPLADEv2 (Formal et al., 2021a; 2022) further improves performance introducing hard-negative mining and distillation. Chen et al. (2021) propose to close the gap with sparse methods on phrase matching and better generalization by combining a dense retriever with a dense lexical model trained to mimic the output of a sparse retriever (BM25). Ma et al. (2021) combine single hybrid vectors and data augmentation via question generation. In Section 3 (Table 2b) we evaluate our search environment and some of the methods above.

The application of large LMs to retrieval, and ranking, presents significant computational costs for which model distillation (Hinton et al., 2015) is one solution, e.g. DistillBERT (Sanh et al., 2019). The generalization capacity of dual encoders have been scrutinized recently in QA and IR tasks (Lewis et al., 2021; Zhan et al., 2022; Ren et al., 2022). Zhan et al. (2022) claims that dense

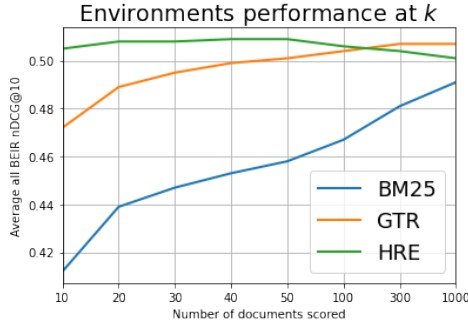

(a) Average nDCG@10 of the BM25, GTR and HRE at different retrieval depths.

| | BEIR subset nDCG@10 | | |
| | ColB. | SPAR. | SPL. |
| --- | --- | --- | --- |
| ColBERTv2 | 0.481 | - | - |
| SPAR | 0.475 | 0.482 | - |
| SPLADE++ | 0.475 | 0.508 | 0.482 |
| HRE, 770M | 0.543 | 0.526 | 0.507 |
| HRE, 11B | 0.529 | 0.530 | 0.507 |

(b) Our HRE, compared, at $k=10$, vs. other dense/sparse methods. The BEIR average score is computed on the subsets of tasks selected by each method. HRE 770M is trained on BM25 top 100 documents, HRE 11B on BM25 top 1000.

Figure 2: Preliminary evaluation of our HRE and benchmark environments on BEIR tasks.

rerankers generalize better than dense retrievers. Ni et al. (2021) suggests that increasing the dual encoder model size increases its ability to generalize. Rosa et al. (2022) argue that large rerankers provide the most effective approach, particularly in zero-shot performance and in combination with a sparse retriever. Their best MonoT5 (Nogueira et al., 2020b) model, a pretrained transformer encoder-decoder finetuned for query-document scoring, yields the state-of-the-art results on 12 out of 18, zero-shot tasks on the BEIR task (Rosa et al., 2022). They observe that in-domain performance is not a good indicator of zero-shot performance. Consequently, they regularize the reranker, trading off in-domain performance and improving zero-shot results.

Another interesting line of research is inspired by large decoder-only language models, where increasing size systematically improves zero-shot performance, as proven by GPT (Brown et al., 2020) and PaLM (Chowdhery et al., 2022). Accordingly, SGPT (Muennighoff, 2022) extends the encoder/decoder-only approach to search to decoder-only modeling, via prompting and finetuning. Also related is the line of work on retrieval-augmented language models (Lewis et al., 2020; Guu et al., 2020) and iterative query reformulation for question answering (Guo et al., 2017; Buck et al., 2018; Qi et al., 2019; 2021; Zhu et al., 2021; Nakano et al., 2021).

## 3    HYBRID RETRIEVAL ENVIRONMENT (HRE) AND BENCHMARKS

A search *environment* is composed of one or more *retrievers* operating over a document collection, whose output is possibly combined, and eventually rescored by a dedicated model, the *reranker*.

### 3.1    RETRIEVERS

We experiment with three types of retrieval methods. The first, BM25, uses Lucene's implementation of BM25[1] as the retriever. This is the setting of (Adolphs et al., 2022). The second environment, GTR, uses GTR-XXL (Ni et al., 2021) as the retriever. The last is a hybrid environment that combines the results of the BM25 and GTR retrievers. We call this a *Hybrid Retrieval Environment* (HRE). After retrieval, HRE simply joins the two $k$-sized results sets, removing duplicates. Thus, for a fixed value of $k$, HRE has available a slightly larger pool of documents, at most $2k$.

### 3.2    THE T5 RERANKER (T5-R)

After retrieval, and, in the case of HRE, the combination step, the top documents are reranked by the environment's reranker, which we refer to as **T5-R**. In contrast to encoder-decoders (Nogueira et al., 2020a) we follow the work of Zhuang et al. (2022) and only train T5-R's encoder and add a classification layer on top of the encoder output for the first token, similar to how BERT (Devlin et al., 2019) is often used for classification.

---

[1]https://lucene.apache.org/.

Instead of using a point-wise classification loss, we use a list-based loss (Jagerman et al., 2022; Zhuang et al., 2022): for each query, we obtain one positive ($y = 1$) and $m - 1$ negative ($y = 0$) documents to which the model assigns scores $\hat{\mathbf{y}} = \hat{y}_1^m$. We use a list-wise softmax cross-entropy loss (Bruch et al., 2019):

$$\ell(\mathbf{y}, \hat{\mathbf{y}}) = \sum_{i=1}^{m} y_i \log \frac{e^{\hat{y}_i}}{\sum_{j=1}^{m} e^{\hat{y}_j}}. \tag{1}$$

We train T5-R on MS MARCO and the output of BM25. We find that a T5-Large trained on the top-100 documents works well on the top results, but a T5-11B model trained on the top-1000 BM25 documents works better in combination with a search agent (Table 2). For HRE we consider the latter reranker. Figure 2 provides a first evaluation of the ranking performance of our environments on the BEIR dataset – see §5 for more details on the task. Figure 2a reports the effect of reranking an increasing number of documents. BM25 provides a baseline performance, and benefits the most from reranking more results with T5-R. GTR starts from a higher point, at $k$=10, and plateaus around $k$=300 with an average nDCG@10 (normalized Discounted Cumulative Gain) of 0.507 on the 19 datasets. Despite its simplicity, HRE is effective at combining the best of BM25 and GTR at small values of $k$. HRE reaches its maximum, 0.509, at $k$=40 but scores 0.508 at $k$=20.

In Figure 2b we situate our environments in a broader context, by comparing zero shot performance against recent dense/sparse combined retrieval proposals: ColBERTv2 (Santhanam et al., 2022), SPAR (Chen et al., 2021) and SPLADE++ (Formal et al., 2022), discussed also in §2. Each of them evaluates on a different subset of the BEIR zero shot tasks, which we select appropriately. HRE produces the best performance by a substantial margin in two configurations. ColBERT and SPLADE do not use a reranker but require more involved training through cross-attention distillation, and rely on token-level retrieval. The best SPAR model needs an additional dense lexical model and relies on a more sophisticated base retriever, Contriever (Izacard et al., 2021). As the results show, a reranker combined with HRE at ($k$=10) provides a simple and effective hybrid search system.

## 4 Hybrid agent Retrieval Environment (HaRE)

A search agent generates a sequence of queries, $q_0, q_1, q_2, \ldots, q_T$, to be passed on, one at a time, to the environment. Here, $q_0$=$q$ is the initial query and $q_T$ is the last one, in what we also call a *search session*. At each step, the environment returns a list of documents $\mathcal{D}_t$ for $q_t$. We also maintain a list, $\mathcal{A}$, of the best $k$ documents found during the whole session

$$\mathcal{A}_t := \{d_i \in \mathcal{D}_t \cup \mathcal{A}_{t-1} : |\{d_j \in \mathcal{D}_t \cup \mathcal{A}_{t-1} : f(q_0, d_i) < f(q_0, d_j)\}| < k\} \tag{2}$$

where $f : (q, d) \mapsto \mathbb{R}$ is the score, $P(y=1|q, d)$, predicted by the T5-R model. When no documents can be retrieved after issuing $q_{t+1}$ or after a maximum number of steps, the search session stops and $\mathcal{A}_t$ is returned as the agent's output. The agent's goal is to generate queries such that the output, $\mathcal{A}_T$, has a high score under a given ranking quality metric, in our case nDCG@10.

### 4.1 Query Refinement Operations

As in (Adolphs et al., 2022), $q_{t+1}$ is obtained from $q_t$ by *augmentation*. That is, either by adding a single term, which will be interpreted by Lucene as a disjunctive keyword and contribute a component to the BM25 score, or including an additional unary search operator. We experiment with the same three unary operators: '+', which limits results to documents that contain a specific term, '-' which excludes results that contain the term, and '$\wedge_i$' which boosts a term weight in the BM25 score by a factor $i \in \mathbb{R}$. We don't limit the operators effect to a specific document *field*, e.g., the content or title, because in the BEIR evaluation there is no such information in the training data (MS MARCO). Formally, a refinement takes the following simplified form:

$$q_{t+1} := q_t \, \Delta q_t, \Delta q_t := [+| - |\wedge_i] \ w_t, w_t \in \Sigma_t \tag{3}$$

where $\Sigma_t$ is the vocabulary of terms present in $\mathcal{A}_t$. This latest condition introduces a *relevance feedback* dimension (Rocchio, 1971). If available, document relevance labels can be used to build an optimal query, e.g., for training. Or, in the absence of human labels, the search results are used for inference purposes – as *pseudo relevance feedback*.

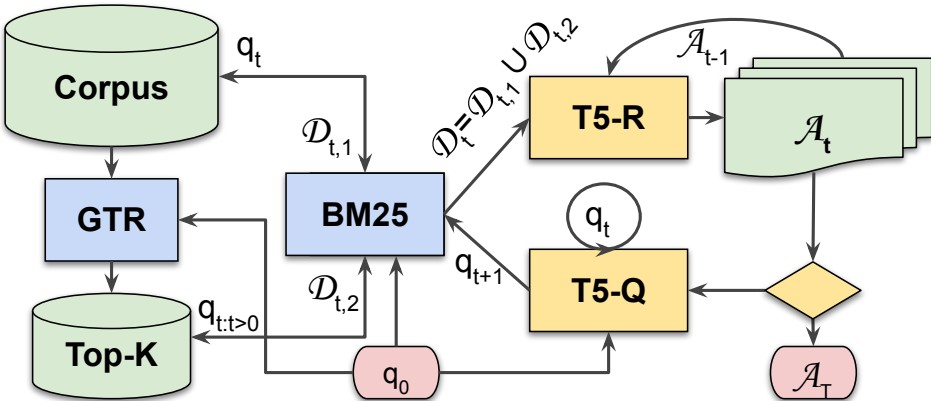

Figure 3: Schematic view of the HARE search agent. The information flows from the input query $q_0$ to the output $\mathcal{A}_T$. In between, retrieval steps (the blue components) and aggregation and refinement steps (the yellow components) alternate in a cycle.

## 4.2 THE T5 QUERY EXPANDER (T5-Q)

A search agent includes an encoder-decoder transformer based on T5 (Raffel et al., 2020) that generates query refinements. We call this component **T5-Q**. At each step, an observation $o_t := (q_t, \mathcal{A}_t)$ is formed by concatenating $q_t$ and $\mathcal{A}_t$, which is a string with a minimal set of structural identifiers. T5-Q takes $o_t$ as input and outputs $\Delta q_t$, allowing the composition of $q_{t+1} = q_t \Delta q_t$, as in Eq. (3).

## 4.3 HARE AND BENCHMARK SEARCH AGENTS

Figure 3 illustrates the HARE search agent. On the first search step only, GTR retrieves the top-1000 documents for $q_0$. These define a sub-collection, Top-$K$, kept frozen through the search session. The top-$k$ documents from Top-$K$, $\mathcal{D}_{0,2}$, are combined with the top-$k$ from BM25, $\mathcal{D}_{0,1}$, also retrieved from the full collection. GTR is not used again. Top-$K$ is further accessed only through BM25, i.e., for $t > 0$. At every step, $t$, the results from the full corpus, $\mathcal{D}_{t,1}$, and those from Top-$K$, $\mathcal{D}_{t,2}$, are joined to form $\mathcal{D}_t$. $\mathcal{D}_t$, in turn, is joined with the current session results $\mathcal{A}_{t-1}$, to form $\mathcal{A}_t$. $\mathcal{A}_t$ is passed to the query expander model, T5-Q, which compiles the observation $o_t = (\mathcal{A}_t, q_t)$, and generates $\Delta q_t$. The new query, $q_{t+1} = q_t \Delta q_t$, is sent to BM25 for another round of search. When the termination condition is met, the agent returns $\mathcal{A}_T$.

Besides HARE we evaluate two simpler search agents, in alignment with the simpler environments, BM25 and GTR. The first agent (BM25) only uses the BM25 components of HARE (the BM25 environment), thus, it has only access to the results $\mathcal{D}_{t,1}$ in Figure 3. Analogously, the second agent (GTR), only uses the GTR components of HARE (the GTR environment), and has access exclusively to the $\mathcal{D}_{t,2}$ results.

## 5 EXPERIMENTS

We run experiments on the zero-shot retrieval evaluation framework of BEIR (Thakur et al., 2021), which includes 19 datasets on 9 domains. Only MS MARCO is used for training and development. Each dataset has its own document collection which is indexed separately. We use the official TREC eval script[2] for our results. Results for the benchmarks are from the corresponding publications.

## 5.1 DATA

To generate training data for **T5-R** we retrieve $k \in \{100, 1000\}$ documents per query for each query in the MS MARCO training set (532,761 questions) using BM25. To make one example list of length

---

[2] https://github.com/usnistgov/trec_eval/archive/refs/heads/master.zip.

$m$, we take a query and one gold document and sample $m-1$ negatives from the top-k documents. We skip queries if no gold document occurs within the top-k documents which removes about 20% of the data.

The training data for **T5-Q** is generated as follows. Synthetic search sessions are simulated from labeled query documents pairs, $(q, d)$, where $d$ is the relevant document for $q$, in the MS MARCO training set. We then use the *Rocchio Session* Algorithm of Adolphs et al. (2022), to search for the optimal expansion. In a nutshell, at step $t$, terms in $\mathcal{A}_t \cap d$ are evaluated as candidates for disjunctive term augmentations, or in combination with '+' and '$\wedge_i$' operators. Conversely, terms in $\mathcal{A}_t - d$ are candidates in combination with '-'. We attempt at most $M$ candidates for each operator using terms in the document set, $\mathcal{A}_t$, ranked by Lucene's IDF score. Starting from $q_0 = q$, a synthetic session is expanded, one step at a time, with the best scoring augmentation. The procedure stops when the nDCG@10 score of $\mathcal{A}_t$ does not improve, or after five steps. We generate two sets of training data: a high-throughput (HT) data, for faster turnaround in development, which sets $M=20$ and yields 120,563 training examples (single steps) from 23% of the questions where improvements were found; and a high-quality (HQ) data using $M=100$ which yields 203,037 training examples from 40% of the questions for which improvements were found. Table 4, in the Appendix, provides an example gold Rocchio session for the query 'what's the difference between c++ and java'. The nDCG score of the HRE results is 0.0, and by applying two refinements ('+language' and '+platform') the score increases, first to 0.6, then to 1.0. In the training Rocchio sessions, the '+' operator is used for 83% of all refinements. The other operators are each used for only 2-3% of all refinements. Although '+' is used for the majority of refinements, when we evaluated the agent's headroom allowing only the '+' operator, the headroom was lower. We also evaluated the agent's headroom allowing only '$\wedge_i$' operators, the result was also worse. The synthetic data is used to finetune T5-Q via Behavioral Cloning, where each step defines an independent generation task.

## 5.2 MODELS

We use the published model checkpoint for GTR-XXL (Ni et al., 2021), as the off-the-shelf dual encoder.[3] For BM25 we use Lucene's implementation with default parameters (k=0.9 and b=0.4).

As detailed in Section 3, the query-document reranker, **T5-R**, is initialized from the encoder of a pretrained T5 model. The encoder output of the first token is fed into a feed-forward layer to generate a score which we use for ranking query-document pairs. The input is structured as follows:

```
query: {query} document: {document}
```

We experimented with several published checkpoints, including T5-large and T5-11B and found the latter to perform better.[4] Note that while T5-11B has 11B parameters we only train roughly half of them (the encoder side). We train with a batch size of 64 and lists of length $m = 32$, yielding an effective batch size of 2048. To limit memory consumption we truncate our inputs to 256 tokens.

The query expander, **T5-Q**, is based on the T5.1.1-XXL model. The input is structured as follows:

```
query: {query} document: {document1}... document: {document10}
```

When examining the training data (§5.1), we found multiple examples of that we consider unlearnable, e.g., involving stop words. As we do not want the search agent to concentrate and overfit on these examples, we employ a simple self-supervised training trick. In each batch, we sort the sequences by their negative log-likelihood (NLL) and we mask out the loss for 50% of the training examples with the highest NLL. Examples can be seen in Table 3 in the Appendix. We are essentially training with only a halved batch size while wasting the other half, but given that the model converges quickly, this technique is sufficient. To further avoid overfitting, we use a small constant learning rate of $3 * 10^{-5}$. We train for 12,000 steps with a batch size of 128 (forward pass before masking), which is equivalent to around 1.5 epochs. The input sequences have a maximum length of 1024 and the maximum target length is set to 32. All other hyperparameters are the T5 defaults.

## 5.3 RESULTS

Table 1 holds the detailed BEIR results. We report the average over all datasets (Average), and minus MS MARCO (Avg. Zero-shot). As benchmarks, we compare with MonoT5, the current SOTA,

---

[3]https://github.com/google-research/t5x_retrieval.
[4]https://github.com/google-research/text-to-text-transfer-transformer.

| Dataset | Benchmarks | | Environments | | | Agents | | | | RS |
|---|---|---|---|---|---|---|---|---|---|---|
| | MonoT5 | SGPT | BM25 | GTR | HRE | BM25 | GTR | RM3 | HARE | BM25 |
| MS MARCO | 0.398 | 0.399 | 0.285 | 0.470 | 0.479 | 0.361 | 0.480 | **0.483** | **0.483** | 0.557 |
| Trec-Covid | 0.794 | **0.873** | 0.579 | 0.537 | 0.666 | 0.778 | 0.703 | 0.744 | 0.765 | 0.921 |
| BioASQ | **0.574** | 0.413 | 0.315 | 0.344 | 0.427 | 0.453 | 0.470 | 0.468 | 0.493 | 0.654 |
| NFCorpus | **0.383** | 0.362 | 0.343 | 0.358 | 0.377 | 0.380 | 0.368 | 0.380 | **0.383** | 0.508 |
| NQ | 0.633 | 0.524 | 0.419 | 0.637 | 0.655 | 0.528 | 0.664 | 0.661 | **0.669** | 0.724 |
| HotpotQA | 0.758 | 0.593 | 0.605 | 0.651 | 0.713 | 0.694 | 0.734 | 0.734 | **0.759** | 0.850 |
| FiQA-2018 | 0.513 | 0.372 | 0.268 | 0.504 | 0.513 | 0.355 | 0.516 | 0.520 | **0.525** | 0.564 |
| Signal-1M | 0.314 | 0.276 | **0.371** | 0.273 | 0.320 | 0.355 | 0.313 | 0.310 | 0.318 | 0.476 |
| Trec-News | 0.472 | **0.481** | 0.300 | 0.368 | 0.394 | 0.353 | 0.368 | 0.420 | 0.406 | 0.521 |
| Robust04 | 0.540 | 0.514 | 0.384 | 0.513 | 0.556 | 0.513 | 0.514 | 0.565 | **0.589** | 0.728 |
| ArguAna | 0.287 | **0.514** | 0.318 | 0.352 | 0.327 | 0.246 | 0.362 | 0.237 | 0.260 | 0.389 |
| Touche-2020 | 0.299 | 0.254 | **0.536** | 0.249 | 0.325 | 0.518 | 0.251 | 0.321 | 0.320 | 0.673 |
| Quora | 0.840 | 0.846 | 0.650 | 0.875 | **0.876** | 0.769 | 0.874 | 0.873 | 0.873 | 0.880 |
| DBPedia | **0.477** | 0.399 | 0.290 | 0.428 | 0.453 | 0.383 | 0.432 | 0.463 | 0.476 | 0.549 |
| SCIDOCS | 0.197 | 0.197 | 0.166 | 0.173 | 0.196 | 0.181 | 0.197 | 0.198 | **0.201** | 0.280 |
| Fever | **0.849** | 0.783 | 0.783 | 0.811 | 0.829 | 0.813 | 0.817 | 0.831 | 0.832 | 0.866 |
| Climate-Fever | 0.280 | **0.305** | 0.222 | 0.282 | 0.296 | 0.258 | 0.287 | 0.300 | 0.300 | 0.408 |
| SciFact | **0.777** | 0.747 | 0.683 | 0.707 | 0.751 | 0.707 | 0.743 | 0.751 | 0.756 | 0.797 |
| CQADupStack | 0.415 | 0.381 | 0.316 | 0.431 | 0.448 | 0.364 | 0.448 | 0.451 | **0.452** | 0.526 |
| #docs reranked | 1000 | - | 10 | 10 | 17.5 | 21.6 | 23.6 | 33.4 | 66.7 | 15.5 |
| Average | 0.516 | 0.485 | 0.412 | 0.472 | 0.505 | 0.474 | 0.502 | 0.511 | **0.519** | 0.625 |
| Avg. Zero-shot | **0.522** | 0.490 | 0.419 | 0.472 | 0.507 | 0.480 | 0.503 | 0.513 | 0.521 | 0.628 |

Table 1: Full results on BEIR. MonoT5 refers to the best-performing system in (Rosa et al., 2022), current SOTA performance holder. MonoT5 reranks the top 1000 documents from BM25 with a cross encoder. SGPT refers to the best performing GPT-style system from (Muennighoff, 2022), SGPT-BE 5.8B. As environments, we evaluate BM25, GTR and HRE (§3). The last four columns report the results of the BM25, GTR and HARE agents (§4.3) including a variant (RM3, based on HRE) that replaces T5-Q with RM3 (Pal et al., 2013). As customary on BEIR evals, we report the all datasets average and without MS MARCO (zero shot only). We also report the number of unique documents scored with the reranker, the average value for agents and HRE. The SGPT model retrieves over the full collection. The last column (RS) reports the performance of the (HQ) Rocchio Session algorithm used to generate the training data when run on all BEIR eval sets. Having access to the labeled document(s), it provides an estimate of the headroom for search agents.

which reranks the top 1000 documents from BM25, with a cross encoder transformer reranker (Rosa et al., 2022). We also report the results of the best performing GPT-style system from (Muennighoff, 2022). SGPT is intriguing because of its unique performance profile (e.g., see on Trec-Covid and Arguana), reinforcing the suggestion that large decoder-only LMs introduce genuinely novel qualitative dimensions to explore. Next, we discuss the results of our BM25, GTR and HRE and corresponding agents. For all our systems, reranking depth is fixed at $k=10$.

All search agents run fixed 5-steps sessions at inference time. They all outperform their environment. One needs to factor in that agents score more documents, because of the multi-step sessions. The BM25 and GTR agents collect on average about 20 documents per session, HARE 66.7. One of the desired features of search agents is deep but efficient exploration, which is observable from the experiments. For the BM25 environment to match the BM25 agent performance (0.474/0.480), one needs to go down to $k \approx 300$, cf. Figure 2a. Overall, the BM25 agent outperforms the BM25 environment by more than 6 points. We highlight that the BM25 agent outperforms also the GTR environment, though not in-domain on MS MARCO – consistently with the findings of Adolphs et al. (2022). The GTR agent outperforms the GTR environment by 3 points, with the GTR environment beginning to perform better than the GTR agent only between $k=50$ and $k=100$.

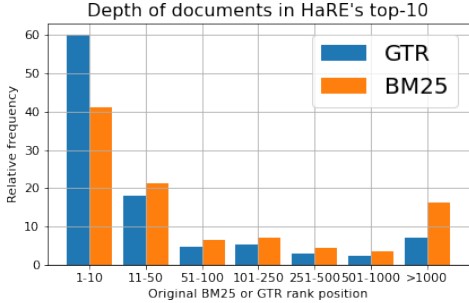
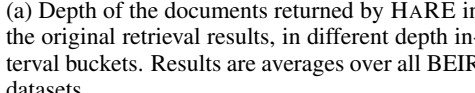
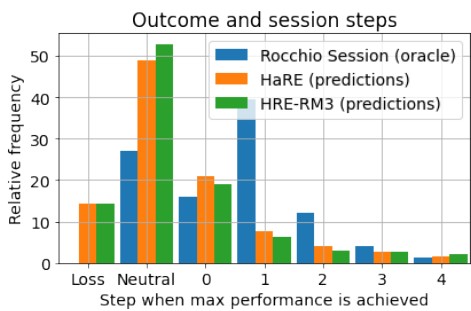

(a) Depth of the documents returned by HARE in the original retrieval results, in different depth interval buckets. Results are averages over all BEIR datasets.

(b) Outcome of Rocchio sessions (oracle) compared to HARE's and RM3's predictions, as a function of the steps required to achieve the maximum score. Averages over all BEIR datasets.

Figure 4: Analysis of BEIR tasks results.

With HRE, performance starts to saturate. However, HARE outperforms HRE by 1.4 nDCG points, scoring on average 66.7 documents vs. the 17.5 of HRE. Note that HRE's maximum performance is 0.509/0.510, at $k = 40$, thus is never near HARE at any retrieval depth. HARE's performance is comparable to MonoT5, the current SOTA: better on all datasets average by 0.3 points and worse on zero-shot only, by 0.1 points. However, HARE scores 15X fewer documents. A conservative estimate shows a consequent 50% latency reduction (cf. §A.1). Furthermore, HARE's in-domain performance (0.483 on MS MARCO) keeps improving and is 8.5 points better than MonoT5. HARE has the best performance on 8 datasets (5 for MonoT5). We also evaluate a variant of the HRE search agent based on RM3 (Jaleel et al., 2004; Pal et al., 2013) a robust pseudo relevance feedback query expansion method (Lv & Zhai, 2009; Miao et al., 2012), which replaces T5-Q as the query expander. At each step, we pick the highest scoring term based on the RM3 score and add the selected term to the previous query with a '+' operator. The RM3 search agent is also effective, it improves over HRE but it does not perform as well as HARE, the reason being that it does not pull in enough new documents (33.4).

The last column in Table 1 reports an oracle headroom estimate. This is obtained by running the same Rocchio Session algorithm used to produce T5-Q's training data (§5.1) at inference time. As the numbers show, there is substantial room for improvement. In the next section we continue with an in-depth analysis and open questions.

### 5.4 QUALITATIVE ANALYSIS

Figure 4a plots the average depth of the documents in HARE's final top-10, over all BEIR tasks in the retrieval results from BM25, and GTR, for the original query. HARE digs deep in the original retrievers rankings, even beyond position 1000: 16.3% of HARE top-10 docs for BM25, and 6.9% for GTR. In this respect, HARE extends the finding of (Adolphs et al., 2022) to neural retrievers.

Figure 4b looks at the outcomes of HARE and HRE-RM3 episodes, compared to oracle sessions. 49% of the time HARE doesn't change the outcome (RM3, 52.8), 14% of the results are worse for both, 21% of the examples are resolved at the initial query for HARE (18.9 for RM3). However, 16% are improved by HARE (14 for RM3) and 8.3% need two or more steps. Table 5, in Appendix, looks in detail at an example of HARE win at step 2: 'what do you use dtp for +software +publishing'. Another, 'when did rosalind franklin take x ray pictures of dna +52 +taken +franklin +photo' needs four steps. A single step one, but particularly transparent semantically is 'what is the age for joining aarp +requirements'. Hence, we find evidence of multistep inference and interpretable actions. Compared to HRE-RM3, HARE explores more documents in fewer steps. This is in part due to RM3 term weighting over-relying on the query. For example, RM3 needs three refinements to solve the query 'what make up the pistil in the female flower', '+pistil +female +stigma', while HARE solves it in one with '+ovary'.

| Environment | T5-R BM25-100 Large | | | T5-R BM25-100 11B | | |
| --- | --- | --- | --- | --- | --- | --- |
| | T5-R | T5-Q Large | T5-Q 11B | T5-R | T5-Q Large | T5-Q 11B |
| BM25 | 0.437 | 0.450 | 0.447 | 0.439 | 0.449 | 0.456 |
| GTR | 0.467 | 0.491 | 0.493 | 0.472 | 0.493 | 0.495 |
| HRE | 0.506 | 0.506 | 0.507 | 0.506 | 0.510 | 0.511 |

Table 2: Average nDCG@10 on BEIR. 'T5-R BM25-100 Large' and 'T5-R BM25-100 11B' are, respectively, T5-Large and T5-v1_1-XXL reranker models trained on the Top 100 BM25 documents from MS MARCO. T5-Q Large and T5-Q 11B are T5-Large and T5-11B agent models trained on data generated via the high-throughput Rocchio Session process (HT, cf. §5.1).

We find that the HARE learns only to use '+', and completely ignores other operators. Part of the problem may be an unbalanced learning task for T5-Q ('+' covers 83% of training). One way to make the task more expressive and balanced would be defining more operators. Lucene, for example, makes available several other operators including proximity, phrase and fuzzy match, and more. More generally, while interpretable, such operators are also rigid in their strict syntactic implementation in traditional search architectures. An interesting direction to explore is that of implementing such operators in a semantic framework, that is via neural networks, combining transparency and flexibility. SPAR's approach to modeling phrase matches (Chen et al., 2021), for instance, is one step in this direction. Another possible limitation is the lack of title information in the training data, as more structure in the data may partition the learning space in more interesting ways. The importance of the document title, for one, is well-known in IR.

In early development phases on MS MARCO we trained several, T5-R and T5-Q, models using different configurations. We first trained T5-R models with the top 100 documents from BM25. We call these models 'T5-R BM25-100'. The T5-R model is used to generate training data for T5-Q, but it is also paired with the trained T5-Q model at inference time. Thus, the T5-R model needs to be robust when faced with documents originating from deeper than at training time. Larger T5-R models seem more robust in this respect, consistently with previous findings (Ni et al., 2021; Rosa et al., 2022). Similarly, larger T5-Q models seem more robust when training on the noisy data generated by the Rocchio Session procedure. Some of those steps are genuinely meaningful, some are spurious actions with little chance of being learnable. Eventually, we settled on the largest models available and trained the T5-R models with the top 1000 documents from BM25. Table 2 provides a sample of these explorations, evaluated on the BEIR dataset as an ablation study.

There are still open questions on how to properly train these models. For instance, it is apparent that our reranker is not as robust as as we would hope for the document depth that is usually explored by agents. Figure 2a clearly shows performance declining at $k>50$. This may, at least partially, explain why we don't squeeze more performance from the existing headroom. A natural option, despite the negative findings of (Adolphs et al., 2022), is joint reinforcement learning of the agent.

## 6 CONCLUSION

In this paper we extended the learning to search (L2S) framework to hybrid environments. In our approach, we simply combine dense and sparse retrievers, in what we call a hybrid retrieval environment (HRE), and leave results aggregation to a reranker which operates efficiently, only on the very top retrieved results. Our experiments show that our search environment, while simple, is competitive with respect to other hybrid proposals based on more complex design, or relying on reranking retrieval results very deeply. Our search agent, HARE, learns to explore the indexed document collection deeply, but nimbly, keeping the number of documents to rescore low. HARE leverages discrete query refinement steps which produce SOTA-level retrieval performance in a competitive zero-shot task, without degrading in-domain performance. Furthermore, we find evidence of effective multi-step inference, and the actions of the search agent are often easy to interpret and intuitive. Overall, we are inclined to conclude that search agents can support the investigation of performant information retrieval systems, capable of generalization. At the same time, they provide plenty of unexplored opportunities, and challenges, on the architectural and learning side.

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

# A    APPENDIX

| query | query expansion (target) | NLL |
|---|---|---|
| what is the age for hitting puberty | around^8 | 75.6 |
| trigeminal definition | affects^6 | 71.2 |
| is rhinitis painful | common^6 | 70.0 |
| how many glasses of water is required a day | every^4 | 67.5 |
| how soon do symptoms show up for hiv | acute^4 | 67.5 |
| the collection film cast | +collection | 0.5 |
| what kind of fossil is made by an imprint? | +fossil | 1.5 |
| who invented corn flakes | +john | 1.7 |
| what does gelastic mean | +laughter | 1.7 |
| what is acha | +hockey | 1.8 |

Table 3: Examples from the evaluation set with highest respectively lowest negative log-likelihood (NLL) using the trained HARE search agent.

## A.1    LATENCY

We estimate latency by measuring the wall clock time of each individual step for HaRE, and MonoT5, running in the same setup; i.e., using the same sub-systems and configurations. For both MonoT5 and HARE we don't include the full collection indexing times which are performed once, and focus on inference.

For MonoT5, we consider only the BM25 retrieval step, plus the T5-R inference to rerank the top 1000 documents returned. We call this $L_{\text{MonoT5}}$. For HARE, we count: BM25 retrieval + GTR retrieval + T5-R inference + 4×(T5-Q inference + BM25 retrieval + BM25 Top-K retrieval + T5-R inference). We call this $L_{\text{HaRE}}$. The resulting ratio is

$$\frac{L_{\text{HaRE}}}{L_{\text{MonoT5}}} = 0.51.$$

Notice that, while we count the T5-R reranking latency in full for each step, in practice the same documents are often retrieved multiple times and caching can be used effectively.

More importantly, for simplicity sake, in the current implementation, we don't create a separate Lucene index for the GTR Top-1K sub-corpus. Instead, to search on GTR's Top-1K corpus, we search on the full collection but restrict the query to the Top-1k documents by means of Lucene's 'id' query operator. Hence, we append 1k document ids to the query, thus constraining retrieval to Top-1K only. This turns out to be by far the slowest step for HARE. A conservative back of the envelope calculation shows that by refactoring the search step by indexing separately Top-1K for each episode and executing two separate BM25 retrieval steps, on the the full and Top-1K collections, would cut overall latency by another 50%.

In general, while more complex, HARE's architecture offer's many options for refactoring and optimization.

## A.2    EXAMPLES

| | Query and Results | Score |
|---|---|---|
| $q_0$ | what's the difference between c++ and java | 0.0 |
| $d_1$ | The difference between C#/Java and C is too big, but the differences between C#/Java and C++ are easier to pick and the most important, other than said updates to the language, are the adoption of a pure OOP approach to programming. | |
| $d_2$ | JavaScript is an interpreted computer programming language. The main difference between Java and JavaScript is that while Java is a programming language, JavaScript is scripting language. It is a prototype-based scripting language that is dynamic, weakly typed, and has first-class functions. However, like Java, JavaScript is heavily influenced by the C programming language. ... | |
| $q_1$ | what's the difference between c++ and java +language | 0.6 |
| $d_1$ | The most important difference is that Java is a memory-safe language, whereas C++ is not. This means that errors in Java programs are detected in defined ways; for example, attempting a bad cast or indexing an array out of bounds results in an exception. | |
| $d_2$ | Java (bottom) is contrasted with C/C++ (top). Unlike C/C++, which is compiled into machine language for a specific hardware platform, Java programs are compiled into an intermediate bytecode language. The bytecode is then compiled into machine language by a runtime program called the Java Virtual Machine (JVM). Unlike C/C++, which is compiled into machine language for a specific hardware platform, Java programs are compiled into an intermediate bytecode language. The bytecode is then compiled into machine language by a runtime program called the Java Virtual Machine (JVM). ... | |
| $q_2$ | what's the difference between c++ and java +language +platform | 1.0 |
| $d_1$ | C++ is an evolution to C. Which was a system programming language. C++ Added many features to the language to make it object oriented. It became the mainstream programming language for that reason. Java is an evolution of C++, with different goals ( cross platform for instance ). It remove some of the features that make C++ so hard to learn. Simplify others and remove others. | |
| $d_2$ | Java (bottom) is contrasted with C/C++ (top). Unlike C/C++, which is compiled into machine language for a specific hardware platform, Java programs are compiled into an intermediate bytecode language. The bytecode is then compiled into machine language by a runtime program called the Java Virtual Machine (JVM). Unlike C/C++, which is compiled into machine language for a specific hardware platform, Java programs are compiled into an intermediate bytecode language. The bytecode is then compiled into machine language by a runtime program called the Java Virtual Machine (JVM). ... | |

Table 4: Example of multistep gold search session that forms the training data for T5-Q. This is one of few examples from MS MARCO where more than one document is annotated as relevant. The first set of results are on topic but too generic, and slightly off the mark: the first document talks also about c#, the second document is about JavaScript. By restricting document to those containing the term 'language' a relevant document is in 2nd position in the results from step 1. Here a new term is discovered, 'platform', which was not present in the results for $q_0$ (it does not occurr in any of the top 10 results, which we omit for simplicity). By further refining the query with +platform, the second-step results contain the two relevant documents at the top and and the session achieves a full score.

| | Query and Results | Score |
|---|---|---|
| $q_0$ | what do you use dtp for | 0.0 |
| $d_1$ | The Dynamic Trunking Protocol (DTP) is a proprietary networking protocol developed by Cisco Systems for the purpose of negotiating trunking on a link between two VLAN-aware switches, and for negotiating the type of trunking encapsulation to be used. It works on Layer 2 of the OSI model. | |
| $d_2$ | DTP (diptheria, tetanus toxoids and pertussis) Vaccine Adsorbed (For Pediatric Use) is a vaccine used for active immunization of children up to age 7 years against diphtheria, tetanus, and pertussis (whooping cough) simultaneously. DTP is available in generic form. | |
| | ... | |
| $d_9$ | Page Layout Software (Generally Known as DTP Software). Since page layout software is commonly known as DTP software, this can lead to some confusion but now you know better. These software programs are the workhorses of DTP and they do exactly what you might think they would in accordance with the name. | |
| | ... | |
| $q_1$ | what do you use dtp for +software | 0.0 |
| $d_1$ | the operating systems main interface screen. desktop publishing. DTP; application software and hardware system that involves mixing text and graphics to produce high quality output for commercial printing using a microcomputer and mouse, scanner, digital cameras, laser or ink jet printer, and dtp software. | |
| $d_2$ | Page Layout Software (Generally Known as DTP Software). Since page layout software is commonly known as DTP software, this can lead to some confusion but now you know better. These software programs are the workhorses of DTP and they do exactly what you might think they would in accordance with the name. | |
| | ... | |
| $q_2$ | what do you use dtp for +software +publishing | 1.0 |
| $d_1$ | Desktop publishing. Desktop publishing (abbreviated DTP) is the creation of documents using page layout skills on a personal computer primarily for print. Desktop publishing software can generate layouts and produce typographic quality text and images comparable to traditional typography and printing. | |
| $d_2$ | Scribus, an open source desktop publishing application. Desktop publishing (abbreviated DTP) is the creation of documents using page layout skills on a personal computer primarily for print. Desktop publishing software can generate layouts and produce typographic quality text and images comparable to traditional typography and printing. | |
| | ... | |

Table 5: Example of multistep search session performed by HARE. The first set of results gets an nDCG score of 0.0, and the top 2 seem clearly wrong. Curiously, none of the top-10 documents mentions the word 'publishing'. HARE first selects the refinement '+software'. The corresponding results, while still scoring 0, lead to the presence of the term 'publishing' which is used by HARE as the next refinement '+publishing', which leads to a full nDCG score on the next round.

