# OpenReview forum: "Zero-Shot Retrieval with Search Agents and Hybrid Environments"
_ICLR.cc/2023/Conference — Submitted to ICLR 2023_

### Official Review · Reviewer_PtJd · 2022-10-25

**Confidence:** 3
**Correctness:** 3
**Technical Novelty And Significance:** 2
**Empirical Novelty And Significance:** 3
**Recommendation:** 5

**Clarity, Quality, Novelty And Reproducibility:**

- The technical novelty of the paper seems limited as mentioned in the weakness section of the review.
- It is unclear in the paper how T5-Q receives the whole set of \mathcal{A_t} as the input.
- [Subjective opinion, minor point] The formulation and terms of environment and agent might mislead the readers to suppose that the paper proposes an RL approach for retrieval. (The authors use the term Behavior Cloning, but it seems like they use the standard LM training objective.)

**Strength And Weaknesses:**

**Strengths**

- The paper is meaningful in that it shows the combination of dense and hybrid retrieval with an iterative query-expansion method is robust on both in-domain and zero-shot retrieval.

**Weaknesses**

- The proposed method is a combination of the methods of previous works [1] and [2], which limits the technical novelty of the paper.
- While the authors emphasize that the proposed system reranks one order of magnitude smaller number of documents compared to the state-of-the-art [1], pointing out that it is time-consuming to cross-encoder reranking of 1000 documents as in [1], it is unclear whether this leads to actual latency improvement of the proposed system because the authors do not report the end-to-end retrieval latency. Since the proposed system uses iterative query refinement (five times) which generates the augmented query based on the previous top-ranked documents, the latency of this step would not be negligible. The authors should report the actual latency in order to show the superiority of the proposed system in terms of time cost.
- While the proposed method achieves near-state-of-the-art zero-shot (BEIR) retrieval performance while achieving much higher in-domain (MS MARCO) retrieval performance, it uses a much more complicated pipeline approach for both training and inference than the state-of-the-art method. The creation of the training data for T5-Q using Rocchio Session would be time-consuming considering that it requires calculating nDCG on multiple (M=100) candidate augmented queries to generate one label for a query. Considering this, the gain in retrieval performance seems insufficient to claim a significant advantage over the method of [1].

**Summary Of The Paper:**

This paper combines the methods proposed in the works of Rosa et al., 2022 [1] and Adolphs et al., 2022 [2] to achieve near-state-of-the-art on the BEIR benchmark with promising in-domain performance. Specifically, as in [1], the authors use the sparse retriever and dense retrieval to retrieve the set of relevant documents which are sorted using a cross-encoder reranker named T5-R, which they call Hybrid Retrieval Environment (HRE). After the initial retrieval, the query is augmented by a query refinement model named T5-Q proposed in [2]. T5-Q is trained to generate an operator with a term, where the term is from the top-ranked documents and the operator is whether to (1) limit the retrieval to the chosen term, (2) exclude the term from the retrieval results, or (3) boost the BM25 term weight of the chosen term. The augmented query is used to run another round of BM25 retrieval, and the newly retrieved documents together with the top-reranked documents serve as the new set of documents to be reranked. The refinement is done iteratively (5 times in the experiments).

**References**

[1] Guilherme Moraes Rosa, Luiz Bonifacio, Vitor Jeronymo, Hugo Abonizio, Marzieh Fadaee, Roberto Lotufo, and Rodrigo Nogueira.  No parameter left behind: How distillation and model size affect zero-shot retrieval. arXiv preprint arXiv:2206.02873, 2022.

[2] Leonard Adolphs, Benjamin B ̈orschinger, Christian Buck, Michelle Chen Huebscher, Massimil- iano Ciaramita, Lasse Espeholt, Thomas Hofmann, Yannic Kilcher, Sascha Rothe, Pier Giuseppe Sessa, and Lierni Sestorain. Boosting search engines with interactive agents. Transactions on Machine Learning Research, 2022. URL https://openreview.net/forum?id=0ZbPmmB61g.

**Summary Of The Review:**

While the paper is meaningful in that it shows the effectiveness of using dense and hybrid retrieval with an iterative query-expansion method, its technical novelty is limited as the proposed method is a combination of the methods proposed in two previous works. Considering the necessity of the complicated training data creation process for T5-Q and the pipeline approach of iterative retrieval-reranking-query refinement, the advantage of the proposed method does not seem significant compared to the simple state-of-the-art method. While the authors emphasize the reduced number of reranked documents as an advantage, whether it actually results in decreased latency remains a question for now.

---

### Official Review · Reviewer_iyX7 · 2022-10-25

**Confidence:** 4
**Clarity, Quality, Novelty And Reproducibility:** The paper is well-written and easy to…
**Correctness:** 3
**Technical Novelty And Significance:** 2
**Empirical Novelty And Significance:** 2
**Recommendation:** 5

**Strength And Weaknesses:**

Strengths:

1. The proposed method sounds reasonable.
2. The proposed method achieved SOTA performance on both zero-shot and in-domain evaluations and keeps the number of documents to rescore low.
3. The paper is easy to follow.

Weaknesses:

1. Considering Adolphs et al., 2022, the contribution of the hybrid retrieval environment appears too marginal, especially the simple combination of sparse retrieval and dense retrieval results, which has been explored by many works.

  [1] Karpukhin, Vladimir, et al. "Dense Passage Retrieval for Open-Domain Question Answering." EMNLP. 2020.

  [2] Sidiropoulos, Georgios, et al. "Combining Lexical and Dense Retrieval for Computationally Efficient Multi-hop Question Answering." Proceedings of the Second Workshop on Simple and Efficient Natural Language Processing. 2021.

  [3] Gangi Reddy, Revanth, et al. "Synthetic Target Domain Supervision for Open Retrieval QA." SIGIR. 2021.

  [4] Chen, Tao, et al. "Out-of-Domain Semantics to the Rescue! Zero-Shot Hybrid Retrieval Models." ECIR. Springer, Cham, 2022.

2. There are many related works on iterative retrieval in the open-domain QA that have not been mentioned and compared, such as:

  [1] Guo, Xiaoxiao, et al. "Learning to query, reason, and answer questions on ambiguous texts." ICLR. 2016.

  [2] Qi, Peng, et al. "Answering Complex Open-domain Questions Through Iterative Query Generation." EMNLP. 2019.

  [3] Qi, Peng, et al. "Answering Open-Domain Questions of Varying Reasoning Steps from Text." EMNLP. 2021.

  [4] Zhu, Yunchang, et al. "Adaptive Information Seeking for Open-Domain Question Answering." EMNLP. 2021.

  [5] Nakano, Reiichiro, et al. "WebGPT: Browser-assisted question-answering with human feedback." arXiv preprint arXiv:2112.09332 (2021).

3. Some comparisons with the baseline experimental results do not seem to be fair, especially in terms of model size, and it is recommended to provide detailed comparisons of parametric quantities. Besides, in Fig. 2(b), as far as I know, ColBERTv2 and SPLADE++ are dense and sparse retrieval, respectively, not hybrid retrieval, and in addition, neither of them has re-ranking, so it is not fair to compare them with HRE. Also, in Table 1, although I believe that iterative agents are better than single-step retrieval environments, the retrieval depth used by the environments is too small to adequately account for this. It is recommended that the retrieval depth used by the environment be set to 5*k or the average number of doc reranked of the agents.



**Summary Of The Paper:**

The paper extends the learning-to-search (L2S) framework to hybrid environments. The paper simply combines two dense and sparse retrievers and a reranker to construct a hybrid retrieval environment (HRE). Based on the hybrid environment HRE, this paper conducts a multi-step retrieval by a search agent HaRE, in which each step involves a reranker to rank the newly top-retrieved and previously collected documents, and a query expander to reformulate a query for the next step. Experimental results show that HRE outperforms its baseline retrievers and some hybrid methods. Besides, HaRE produces state-of-the-art performance on both zero-shot and in-domain evaluations and keeps the number of documents to rescore low.

**Summary Of The Review:**

The proposed method is interesting, but has limited novelty, is similar to many missing related works, and unfair comparisons with baselines do not fully demonstrate its effectiveness.

1. Related work on query expansion and pseudo-relevant feedback is not discussed.
2. In Section 4.3, why GTR is used only one time? And after the first step, it is better to explain why the top 1000 documents from GTR and the entire corpus are simultaneously retrieved by BM25.
3. Minor Issues
  a. The third paragraph of the introduction,  “implicit relevance feedback” -> “pseudo relevance feedback”?
  b. No caption in Figure 1
  c. Equation 2 is hard to understand

---

### Official Review · Reviewer_ettj · 2022-11-04

**Confidence:** 4
**Correctness:** 3
**Technical Novelty And Significance:** 2
**Empirical Novelty And Significance:** 2
**Recommendation:** 3

**Clarity, Quality, Novelty And Reproducibility:**

Some comments/questions

* Figure 1: caption?
* List-wise softmax cross-entropy: isn’t this just contrastive learning (infoNCE) loss?
* (Figure 2) Why does the hybrid environment even start to drop after k=50, while BM25 and GTR keeps increasing?
* Figure 3 is very confusing… just reading the text is much clearer than reading this figure.
* Since L2S is such an important baseline, the authors should mark it in the main experiment table. Based on my understanding, agents-bm25 is L2S?


**Strength And Weaknesses:**

## Strength

The authors present a finding that using the combination of dense and sparse agents in a retrieve-then-rerank setting leads to much better performance than a single agent (with fewer than 100 retrieved documents). The final model (hybrid search environment + query refinement) leads to a state-of-the-art model. The ablation study and analysis are comprehensive, showing the strength of the proposed hybrid environment.

## Weakness

If my understanding is correct, the main contribution of the paper is to propose using sparse+dense retrievers +reranking at the first step. Though I don’t think previous works have reported this before, either combing sparse and dense retrievers [1] or using rerankers is not new.

The authors also sell the hybrid environment as it does not need to retrieve hundreds or thousands of documents for reranking, compared to state-of-the-art models like MonoT5. However, it is not fair to only compare the document numbers — first, sparse and dense retrievers have very different efficiencies; second, the proposed model has a multi-step question refinement procedure, which makes it much slower. To truly show the proposed model’s strength, the authors should really show the retrieval time per query of each model.


[1] Ma et al., 2021. A Replication Study of Dense Passage Retriever.

**Summary Of The Paper:**

Adolphs et al., 2022 propose to train a question refinement component as a search agent — a sparse retriever (BM25) first returns a list of results, and then the question refiner edits the original question (add/remove keyword or changes BM25 weight) to get a list of new results, mimicking human agent behavior.

This paper is an extension of Adolphs et al., by (1) using the combination of a dense retriever (GTR) and a sparse retriever (BM25) for the initial list and (2) using a cross-encoder for reranking. The experiment results show that the hybrid retriever is much better than just using either GTR or BM25 with the reranker.

Then the authors conduct comprehensive experiments towards a series baseline, including the state-of-the-art MonoT5. The proposed HARE is better than other models using different environments or agents, and it achieved comparable performance to MonoT5. The authors also conduct some analysis on the agent behavior; for example, they found that HARE learns to use “adding keywords” the most.

**Summary Of The Review:**

If my understanding is correct, the main contribution lies in the use of "hybrid environment" -- combing dense and sparse retrievers and rerankers. Also, there are some flaws in the experiments -- the authors should really compare the retrieval time of different models instead of the number of retrieved documents. Thus I am leaning towards rejection.

---

### Decision · Program_Chairs · 2023-01-20

**Decision:**

Reject

**Justification For Why Not Higher Score:**

All the reviewers have similar concerns about the contributions of the work, so I think the paper should be rejected in its current form.

**Justification For Why Not Lower Score:**

N/A

**Metareview: Summary, Strengths And Weaknesses:**

This paper proposes an iterative query refinement solution to retrieval, which is based on both dense and sparse retrieval to retrieve relevant documents, and a cross-encoder re-ranker to sort them.  The approach achieves strong results on both in-domain and zero-shot retrieval evaluation.

All the reviewers reached a similar conclusion that 1) the contribution compared to prior work (Adolphs et al., 2022) is small (hence concerns about novelty of the work); 2) an extensive discussion of query expansion, iterative retrieval, and pseudo-relevance feedback is needed; 3) given the approach requires iterative retrieval and re-ranker, the latency needs to be discussed thoroughly, although the authors added a discussion during the response phase; 4) it adds a lot of complexity to the approach, which questions whether the improvements are significant or not.

Overall, given all the above concerns, I can’t recommend the acceptance in the current form.